# From the Metabolic Effects and Mechanism of Monovalent Cation Transport to the Actual Measurement of the Plasma Membrane Potential in Yeast

**DOI:** 10.3390/jof11070522

**Published:** 2025-07-15

**Authors:** Antonio Peña, Norma Silvia Sánchez, Martha Calahorra

**Affiliations:** Departamento de Genética Molecular, Instituto de Fisiología Celular, Universidad Nacional Autónoma de México, Circuito Exterior s/n, Ciudad Universitaria, Mexico City 04510, Mexico

**Keywords:** yeast, yeast membrane potential, membrane transport, internal pH, monovalent cations

## Abstract

The effects of potassium (K^+^) on yeast metabolism were documented as early as 1940. Studies proposing a mechanism for its transport started in 1950, and in 1953, a mechanism for the stimulation of fermentation was suggested. However, it was not until the 1970s that both mechanisms were clarified in Mexico, and the actual internal pH of the cells was measured. The presence of an H^+^-ATPase that generates an electric plasma membrane difference (PMP), which is used by specific transporters to facilitate the influx of K^+^ and other cations into the cells, was discovered. For years, many efforts were made to estimate and measure the value of the PMP; the obtained results were variable and erratic. In the 1980s, a methodology was developed to estimate the plasma membrane potential by following the fluorescence changes in the DiSC_3_(3) dye and measuring its accumulation, which provided actual but inaccurate values. Similar values were obtained by measuring the accumulation of tetraphenylphosphonium. The most reliable method of measuring the actual values of the plasma membrane potential was only recently devised using the also fluorescent dye thioflavin T. This review presents the attempts and outcomes of these experiments necessary to clarify the results reported by different research groups. Innovative research with Genetically Encoded Voltage Indicators (GEVIs) is also included.

## 1. Introduction

Understanding the plasma membrane potential (PMP) in yeast is important, because it is a fundamental cellular process that underpins many vital functions, including nutrient uptake, ion balance, stress response, and signal transduction. *Saccharomyces cerevisiae* is an excellent model organism for studying these basic eukaryotic cellular mechanisms due to its ease of genetic manipulation and well-understood biology.

Reviewing methods to determine PMP in yeast is crucial because it helps us navigate the diverse and evolving techniques used to measure this physiological parameter. Each method, whether traditional electrode-based (not applicable in yeast as will be explained), fluorescent dye-based, or genetically encoded voltage indicator-based, has its strengths and limitations. By comparing these approaches, highlighting potential pitfalls, and establishing best practices, we can ensure researchers obtain accurate and reliable results.

This has significant practical implications. It is critical for optimizing industrial processes that rely on yeast, such as biofuel and pharmaceutical production. It also deepens our knowledge of ion transport systems, aids in drug discovery by providing a platform for studying membrane proteins, and sheds light on cell death pathways. In this review, we synthesize the available information on PMP measurement methodologies and historical context for researchers interested in this important physiological function of yeast and perhaps the possible application of this learning could be applicable to other cell types in the future.

## 2. Historical Background

In 1940 Pulver and Verzár, discovered that adding potassium (K^+^) to yeast cells, increased glucose consumption [1]. This led to a series of studies, the first of which, by Conway and Brady in 1947 [2], proposed a simple K^+^-H^+^ exchange. It was also found that by adding K^+^, the internal pH of the cells increased [3], implying an extrusion of protons in an exchange for K^+^ entering the cell, which led to the proposal of a redox mechanism for the uptake of K^+^ by the same group [4,5]. On the other hand, Rothstein and Demis (1953) analyzed the stimulation of fermentation by K^+^ and proposed a mechanism in which the uptake of the monovalent cation, taken up by the cells, possibly stimulated the activity of one or more enzymes of the glycolytic pathway [6]. Furthermore, they also defined the kinetic constants and selectivity for the uptake of monovalent cations, establishing a clear preference for K^+^ [7].

### 2.1. From the Effects of K^+^ to Its Transport Mechanism

Initial studies were performed to define the mechanism of the K^+^ stimulation of fermentation, without conclusive results [8]. It was later that Peña et al. found that the addition of K^+^ produced a rapid decrease in the ATP levels, accompanied by an increase in ADP and phosphate [9]. Inspired by the work of Mitchell in 1960 on the mitochondrial H^+^-ATPase [10], the mechanism of Na^+^ and K^+^ transport in muscle [11], the gastric acid production [12], as well as that by Slayman and Slayman (1970) [13], who described an ATPase in the plasma membrane of *Neurospora crassa*, capable of exchanging hydrogen ions for K^+^, led to propose the existence of a H^+^-ATPase in yeast that generates a plasma membrane potential (PMP), negative inside the cells, which is capable of driving the uptake of K^+^ by another specific transporter [14], but at high pH, it can pump protons out independently of the presence of a monovalent cation. Furthermore, it was proposed by Gaber and collaborators that the electrochemical gradient formed by this H^+^-ATPase also drives the transport of other monovalent cations and some nutrients [15].

In 1986, the group led by Ramón Serrano not only isolated but also described the structure of the plasma membrane H^+^-ATPase (Pma1p) which was found to be similar to the muscle Na^+^-K^+^ and Ca^2+^-ATPases [16]. In 1984, Rodríguez, Navarro, and Ramos found that there were not one but two distinct K^+^ carriers in yeast, which were designated Trk1p and Trk2p [17]. These strains were donated to Gerald Fink’s laboratory, where they performed genetic experiments and characterized the independence of the high affinity (Trk1p) and low affinity (Trk2p) transporters. It was also demonstrated that they were encoded by an unlinked loci to the H^+^-ATPase *PMA1* [18,19,20]. A similar mechanism was also described for the first time in another yeast, *Schizosaccharomyces pombe* [21].

The proposed scheme for the monovalent cation transport in yeast was then as follows:
The plasma membrane contains a H^+^-ATPase (Pma1p) which generates an electric potential difference by pumping protons out, resulting in a negative potential inside the membrane.Two transporters (Trk1p and Trk2p), with different affinities driven by the PMP, are responsible for the uptake of monovalent cations, thereby maintaining the electrical and ionic homeostasis in the cell.In addition, there is an outwardly rectifying K^+^ channel (Tok1p) that is activated under conditions of plasma membrane depolarization [22,23,24,25] (Figure 1).


### 2.2. Regulation of the Internal pH of Yeast

The activity of the plasma membrane H^+^-ATPase, which is involved in the uptake of different cations by generating the PMP, increased with their addition. This stimulation was found to be dependent on the affinity of the cations in question and their subsequent transport. The activity of this H^+^-ATPase is regulated not only by the PMP but also by the external pH. At a high pH of the medium, it becomes active independently of the presence of cations, enabling yeast to acidify the medium [9].

This general idea led to the confirmation of previously described changes in the internal pH of the cells [3]. It was found that both increasing the pH of the medium and adding K^+^ both produced an increase in the internal pH of the cells. However, it was also found that an increase in the cells’ internal pH occurred at a high medium pH (7.0 or higher), even in the absence of K^+^ [9,27].

## 3. Estimating the Value of the Plasma Membrane Potential (PMP)

Attempts to measure the plasma membrane potential (PMP) were initially performed by the accumulation of organic cations. The initial studies began with the measurement of the accumulation of tetraphenylphosphonium via the Nernst equation [28,29,30,31]. Subsequently, the utilization of fluorescent dyes was adopted for this purpose.

### 3.1. The Estimation of the Yeast PMP by Using Fluorescent Dyes

Ethidium Bromide(EtBr)

Following previous work on the use of fluorescent cationic dyes to estimate the PMP of mitochondria, red blood cells, or membrane bilayers [32,33,34,35], our group started by using ethidium bromide (EtBr) in the yeast *S. cerevisiae* [36]. This dye, at low concentrations, contrary to the findings of the aforementioned authors, increased its fluorescence when starved yeast cells were energized by the addition of glucose. A decrease in fluorescence was expected upon the addition of K^+^, but no changes were detected, because, as demonstrated later, ethidium is a competitive inhibitor of K^+^ uptake [37].

DiSC3(3)

A variety of fluorescent dyes started being used for the estimation or supposedly measuring the PMP of cells, including yeasts; among them, one of the most popular was the fluorescent dye DiSC_3_(5). We decided to acquire it; however, the administration of our Institute ordered the also fluorescent DiSC_3_(3). While expecting them to correct their error, we decided to evaluate it and obtained surprisingly positive results [38]. Using *S. cerevisiae* cells, starved for 24 h with this dye, used at low concentrations (200 nM), when added to the cells in the presence of glucose, the following results were obtained:

(a)An increase in fluorescence upon the addition of the dye;(b)After three or four minutes, another slight increase took place, coincident with the exhaustion of oxygen;(c)As expected, the addition of a small concentration of H_2_O_2_, which, by the action of the catalase, re-established the oxygen levels in the sample buffer, resulted in the return of the fluorescence to the previous level;(d)The addition of a low concentration of the uncoupler carbonyl cyanide m-chlorophenylhydrazone (CCCP) (10 µM) resulted in a much larger increase in fluorescence, and finally;(e)The addition of a rather low concentration of KCl (5 to 10 mM) produced a significant decrease, which was not observed by adding NaCl (Figure 2). Moreover, the addition of KCl after that of NaCl still resulted in the fluorescence decrease [38].

These results were interpreted as follows:

(a)The fluorescence changes in starved cells required the addition of glucose to provide energy, and the dye was taken up but mostly accumulated in the mitochondria, where due to the negative internal membrane potential of these organelles and the high concentration reached, its fluorescence was partially quenched.(b)The slight increase in fluorescence upon oxygen exhaustion was due to an efflux of the dye from the mitochondria to the cytoplasm, where, owed to a higher relative volume, fluorescence increased, but not completely, because the mitochondria could still be partially energized by the ATP produced by fermentation.(c)The former was confirmed by the fluorescence diminution after the addition of hydrogen peroxide.(d)The rather high increase in the fluorescence level after adding CCCP was due to the efflux of the dye from the mitochondria to the cytoplasm, where it increases due to its distribution into a larger volume, the cytoplasm. Here it is important to note that a concentration of CCCP low enough to uncouple mitochondria, but not the plasma membrane, was used.(e)Finally, the decrease after the addition of KCl was due to the efflux of the dye, owing to the also partial decrease in the PMP resulting from the uptake of K^+^.

In all experiments, a low concentration of CaCl_2_, later substituted by BaCl_2_ (10 µM) was added to eliminate changes due to the adhesion of the dye to the negative surface of the cells [38].

The assumed actual PMP values were calculated by the accumulation of the dye DiSC_3_(3), also using the Nernst equation [38]. The values appeared correct, but several factors to be considered in the accumulation experiments were omitted, as described below in the analysis of the experiments and the calculation of the actual PMP values. It is particularly important to note that this cyanine is extremely hydrophobic, which causes it to accumulate not only by the PMP but also binds significantly to hydrophobic sites within the cell. This is something that must be considered with all molecules used for this purpose.

Another method by analyzing the fluorescence changes of DiSC_3_(3) was developed by other authors by following the spectral changes in consequence of its uptake and binding to the internal components of the cells [39,40]. This approach was originally used with murine hematopoietic cells [39] and subsequently with yeast [41]. However, what they estimated were the changes in fluorescence maxima due to the accumulation of the dye by the mitochondria. This explains why they also found that very low concentrations of an uncoupler resulted in a close to zero value of the PMP (see ahead). Indeed, their data demonstrated that low concentrations of CCCP or FCCP resulted in the complete absence of spectral changes, not considering that low concentrations of uncouplers reduce the mitochondrial membrane potential but have no effect on the plasma membrane potential. This has been demonstrated indirectly, but unequivocally, by the observation that low concentrations of an uncoupler stimulate respiration in yeast cells, while much higher concentrations are required to inhibit K^+^ transport, which depends on the PMP [42]. The spectral changes in the cyanine DiSC_3_(3) were posited as true indicators of the changes in the PMP in yeast. Although the actual absolute values were not presented, these experiments, which reported only variations to the PMP, finally represent a significant contribution to the topic [41,43].

#### Changes in Fluorescence Depend on the Concentration of the Dyes

When studying the use of ethidium bromide in mitochondria, Gitler et al. in 1969 [32] observed that its fluorescence decreases when added to energized mitochondria, but when Peña and Ramírez in 1975 used it in yeast at low concentrations, an increase in fluorescence levels was shown when glucose was added [44]. These findings highlight that the concentration of the dye also has a crucial role in understanding the fluorescence changes associated with the PMP.

Although most authors using DiSC_3_(5) reported that its fluorescence decreased when the PMP increased, using low concentrations of this dye shows similar, although less clear results than those of DiSC_3_(3), as shown in Figure 3.

Recent research using DiSC_3_(3) molecules to estimate the PMP

Recently, several reports have emerged regarding the estimation of yeast membrane potential in various areas of research. Many of these studies still use the cyanine DiSC_3_(3) to record fluorescence spectra and report the 580/560 nm ratio. For example, in 2020, Hou et al. analyzed whether small non-protein molecules could harness the membrane potential to concentrate potassium in a *trk1Δtrk2Δ* strain. They showed that amphotericin B could do so [45]. Kodedová and colleagues in 2025 studied the relative membrane potential following the replacement of native ergosterol with alternative sterols in *S. cerevisiae* to search for changes in the physiological parameters of the plasma membrane [46]. 

This cyanine is useful not only with *S. cerevisiae*; in 2022, Vázquez-Carrada et al., working with *Ustilago maydis*, a basidiomycete that infects corn and teosinte, described that this fungus has two plasma membrane H^+^-ATPases (PMA1p and PMA2p) and that by constructing the corresponding mutants, they concluded that conserving one of them is sufficient to ensure cell growth, acidification of the external medium, intracellular pH regulation, and generation of the membrane potential [47].

Fluorescent dyes different from DiSC_3_(3)

Current studies have also used fluorescent monitors, as did Biduik and colleagues, who estimated the PMP of yeast cells under conditions of high glucose concentration, neutral pH or 150 mM KCl to evaluate cell necrosis. These authors utilized DiOC_2_(3) as a fluorescent monitor in a plate reader, using 485 nm wavelength for excitation and recording emission at 625 nm. They took the precaution of using a *rho0* strain to avoid mitochondria interfering. With this technique, they concluded that the PMP plays a predominant role in regulating yeast necrosis [48].

In 2024, Zhu et al. investigated the role of *PMP3* in *Candida albicans*. Pmp3p is a small and conserved plasma membrane protein found in many organisms, such as bacteria, yeasts, dinoflagellates, and plants, but not in mammals. In *S. cerevisiae*, Pmp3p is highly abundant in the plasma membrane and is important for maintaining membrane potential. By measuring the intracellular DiBAC_4_(3) fluorescence intensity, they found that the absence of Pmp3p or Pmp5p leads to depolarization of the plasma membrane, finding a relationship between *PMP3* expression and ketoconazole resistance in *C. albicans* [49].

Souza-Guerrerio et al. most recently measured plasma and mitochondrial membrane potential dynamics upon electrical stimulation using the model organism *S. cerevisiae*, considering that membrane potential is a useful marker for antimicrobial susceptibility testing (AST) using the fluorescent indicators DiBAC_4_(3), Thioflavin T (ThT), and tetramethylrhodamine (TMRM). They demonstrated that proliferative cells exhibit hyperpolarization of the PMP, whereas inhibited cells do not. This suggests that BeAST (bioelectrical AST) can be used to assess the efficacy of antimicrobials, optimize treatments, and facilitate the screening of new antimicrobials [50].

Thioflavin T (ThT)

Special mention should be made of ThT, which has recently been used for PMP estimation and actual measurement, and whose characterization is described below.

### 3.2. The Actual Measurement of the Yeast PMP with Thioflavin T (ThT), an Old Dye

As already mentioned, many attempts were made to measure the PMP of yeast cells, which, of course, due to their size and the presence of the cell wall, makes the use of microelectrodes impossible. As described, values obtained through the accumulation of tetraphenylphosphonium gave variable results, even for our group trying to measure it by the accumulation of the cyanine DiSC_3_(3) [38].

Along the experiments involving different dyes, we evaluated the fluorescence changes in a dye from an old Pfaltz and Bauer flask labeled “acridine yellow”, which curiously enough was actually yellow. The results were comparable to those obtained with DiSC_3_(3) as previously described. However, higher concentrations were required, in the range from 25 to 50 µM [51,52]. It was decided to obtain new dyes, one from Pfaltz and Bauer, one from Sigma, and one from Biotium. The new dyes were completely different from the first one, fine powders displaying a dark red tone; so, it was necessary to define what was the real structure of the old “acridine yellow”. Finally, the old dye was identified as Thioflavin T (ThT). New experiments were performed with the new (and fully identified) thioflavin T giving promising results as reported in Peña et al., in 2023 [26].

By following the fluorescence of the cells over a time course, the results resemble those of the cyanine DiSC_3_(3), with the same changes due to the effectors and the same interpretations as above (Figure 4). Other techniques to follow the fluorescence of ThT such as microscopy, flow cytometry, and the use of a multi-well plate reader, have validated these results [26].

The most recent findings, trying to calculate the PMP values, were obtained by measuring the accumulation of thioflavin T [26,53]. Thioflavin T has the advantage over DiSC_3_(3) of being around 17 times less hydrophobic, as measured by its distribution in dichloromethane-water; this allows more accurate results of its incorporation due to the PMP, with a much lower contribution by its simple binding to the hydrophobic components of the cell.

In these experiments, the following factors that had not been considered in previous studies, both ours and from other groups, were considered:

Driven by the PMP, the dye enters the cells, but once inside, it is accumulated by the mitochondria, where due to the high concentration reached, its fluorescence is greatly quenched; this accumulation can be reversed by the addition of a low concentration (10 µM) of CCCP that uncouples the membrane potential of the organelles without affecting the PMP of the cell [42] (Figure 5, panels A and B). A low concentration of KCl (5–10 mM) is capable of depolarizing the cellular plasma membrane, resulting in the release of the dye, lowering the total fluorescence in the sample (Figure 5, panel C). This had been previously seen also for the cyanine DiSC_3_(3) [38].The microscopic images of the cells showed that the dye did not enter the vacuole; therefore, it was necessary to obtain the actual value of the cytoplasm volume, which was calculated from the total volume of the cells obtained with ^14^C-inulin and with the image analysis of the cells, subtracting the vacuolar from the total volume. This gave the actual concentration of the dye in the cytoplasm in the presence of the uncoupler.The accumulation of this dye is not only attributable to the PMP; part of it is due to its hydrophobic and cationic nature, which results from its binding to the internal molecules of the cell. To avoid dragging this error into the PMP calculation by means of the Nernst equation, chitosan was used to permeabilize the cells and allow all the ThT to leave the cell. Any remaining ThT inside is bound to hydrophobic and/or anionic molecules. By adding a high concentration of KCl (200 mM), the ThT bound to anionic molecules is displaced, leaving only that bound due to its hydrophobic character. Consequently, the dye bound solely due to its cationic nature is calculated. Subsequently, the actual concentration of ThT within the cytoplasm can be obtained by subtracting these values from the raw ones. The actual concentration of the dye inside the cytoplasm can be obtained after the cells are incubated with glucose plus CCCP, in the absence, as well as in the presence of K+. With the corrected concentration values in the presence of CCCP, both in the absence and in the presence of KCl, the actual values of the PMP can be obtained by utilizing the Nernst equation. A report with different strains further confirmed the validity of the procedure employed [26].However, to calculate the values of the PMP, it is necessary that the strains used present a sensitivity of the plasma membrane potential lower than that of the mitochondria. When this sensitivity is similar, low concentrations of an uncoupler affect both the mitochondrial and plasma membrane electric potentials, turning uncertain the values of the PMP. Such is the case of *Debaryomyces hansenii, Meyerozyma guilliermondii*, and *Rhodotorula mucilaginosa* in which only approximate values can be obtained with this method [26].

The schematic representation in Figure 5 to visualize dye localization can also be applied to the DiSC_3_(3) fluorescence changes in Figure 2. Recently, the effect of pH on the PMP in yeast has been reported using the ThT method. The results showed that cells at pH 4.0 maintain a lower PMP, but still close to that observed at pH 6.0 and 7.0 as expected. Cells must cope with different stressors, in this case pH, and must maintain a balanced PMP to survive, as the authors explained. This is another validation for our method [53].

### 3.3. A New and Promising Option to Estimate PMP: The Genetically Encoded Voltage Indicator (GEVI) Proteins

Interestingly, a new approach is to use genetically encoded voltage indicator (GEVI) proteins, which are employed in neuroscience to monitor changes in yeast membrane potential. These are fluorescent proteins coupled to a voltage sensor moiety acting as an internal probe that detects the voltage change through a voltage-driven proton transport process that eventually results in fluorescence emission. In 2020, Limapichat and colleagues selected two GEVI proteins from each of the two distinctive GEVI protein families to express them in S. cerevisiae BY4742 cells: ArcLight and Accelerated Sensor of Action Potential (ASAP) from the Class I VSD family, and MacQ-mCitrine and QuasAr2-Citrine from the Class II opsin family. They characterized these proteins to search for the expression and the correct insertion in the plasma membrane. The fluorescence emission intensity obtained would reflect the amplitude of PMP. In an elegantly designed project, they concluded that only ArcLight and ASAP proteins could be expressed and transported to the plasma membrane, concluding that ASAP is the more suitable GEVI protein for reporting yeast PMP changes since ArcLight is also sensitive to internal pH changes. This ASAP-based voltage reporter enables real-time optical monitoring of yeast PMP and offers valuable insight into optimizing industrial growth conditions and studying ion transport [54]. The same group later reported an application of their method; they monitored the PMP during hyperosmotic stress obtaining differential responses in membrane electrophysiology towards different osmotic stress conditions, during an early stage of yeast stress adaptation, validating their method [55].

Nevertheless, it is important to emphasize that although any of the previously described methods can estimate the PMP by fluorescence, none of the reports to date have demonstrated that the PMP can be measured in millivolts, as is the case with Thioflavin T [26].

## 4. Conclusions

Yeast cells transport K^+^ and other cations due to the activity of a membrane H^+^-ATPase, which generates a PMP that drives them inside through specific carriers.

The simple measurement of the accumulation of different hydrophobic cationic molecules provides an approximation but not the precise values of yeast PMP.

The plasma membrane electric potential of the yeast plasma membrane (PMP) and its changes under different conditions can be estimated by following the fluorescence of various dyes.

Currently, the most well-described method to measure the actual values of yeast PMP is by following the accumulation of thioflavin T but applying the necessary correction factors.

The method, however, requires a lower sensitivity of the plasma membrane proton gradient to the uncoupler used, which is more evident with *S. cerevisiae*.

It is important to note that when measuring the PMP by the accumulation of a dye, its hydrophobicity must be taken into account,, since the more hydrophobic a monitor is, the greater its binding to the internal components of the cell, making the results uncertain.

Future directions for measuring plasma membrane potential include expanding the use of GEVIs with improved sensitivity in diverse cell types and live organisms. Their applications in industrial biotechnology, plant stress responses, and drug screening are becoming increasingly relevant. Integrating GEVIs with omics and AI tools could improve diagnostics and real-time cellular monitoring.

## 5. An Additional Consideration

This is one case of how research, mainly in yeast, started in Mexico; however, other cases deserve special mention.

One is the work developed by José Ruiz Herrera. His work was centered mainly in the study of the fungal cell wall and membrane, making a great contribution to this topic in mycological research [56,57].

Another one is that of Aurora Brunner Liebshard, who began in our Institute the study of Molecular Biology, starting with research on the K^+^ transport in *Kluyveromyces lactis*. One of her first publications was the characterization of a mutant resistant to EtBr and defective in K^+^ transport; the degree of EtBr resistance correlated with the extent of cation uptake impairment. This suggests a shared underlying mechanism or a common pathway affected by EtBr resistance [37].

Other authors, like Juan Pablo Pardo, a former Ph. D. student of Antonio Peña, have also made important contributions to the study of different aspects of yeast physiology, including PMP, internal pH, and more [47,58,59].

### Tributes

To Aurora Brunner Liebshard (died in 1994) and José Ruiz Herrera (died in 2023). We honor their memory with this review.

## Figures and Tables

**Figure 1 jof-11-00522-f001:**
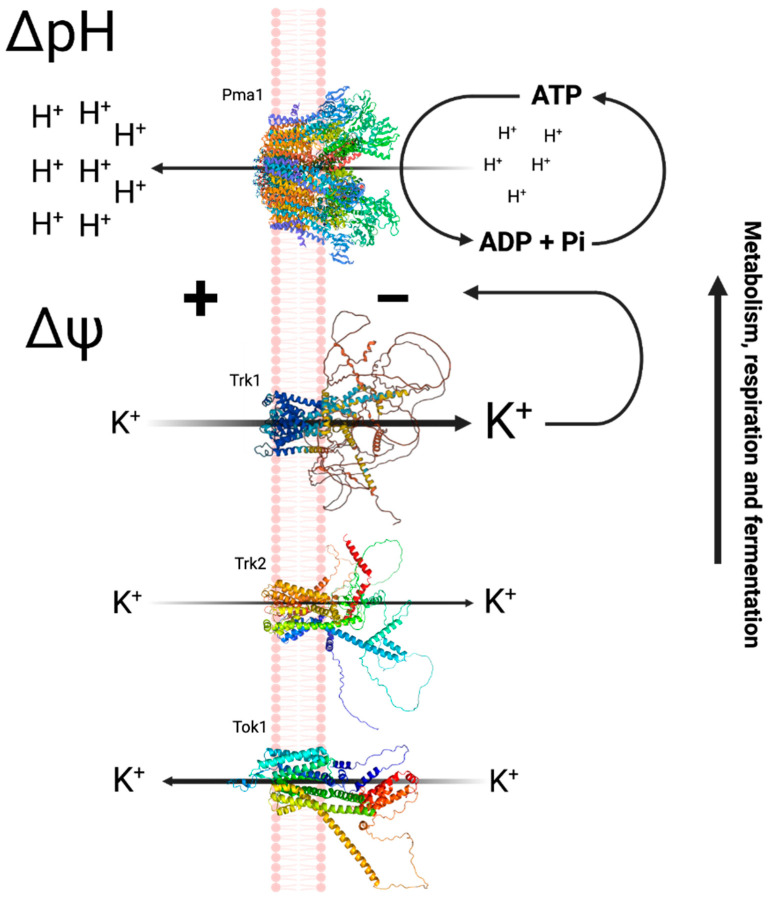
Mechanism of acidification and K^+^ transport in *Saccharomyces cerevisiae*. Pma1p acidifies the external medium by pumping out protons, generating a pH difference (∆pH) with the interior of the cell and an electric potential difference (∆Ψ), that is used to internalize K^+^ ions by Trk1p and Trk2p. Figure from [26].

**Figure 2 jof-11-00522-f002:**
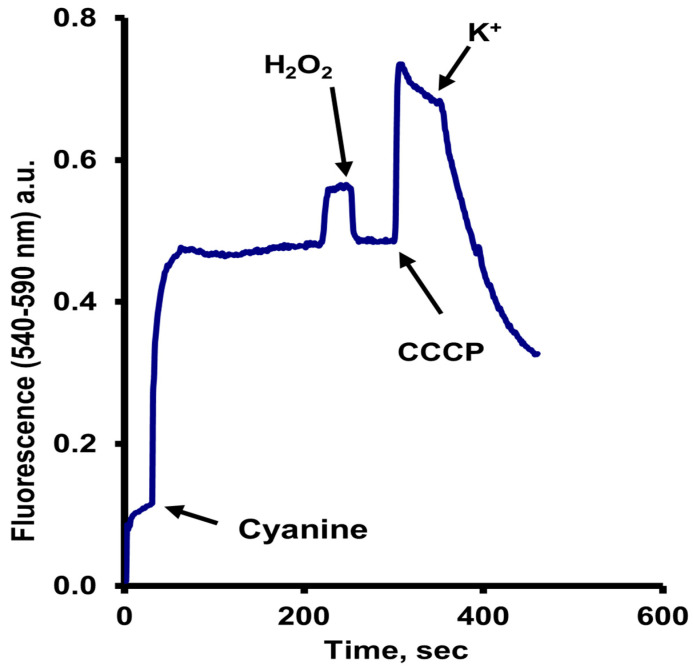
Fluorescence changes of the cyanine DiSC_3_(3) added to starved *S. cerevisiae* cells. The changes in the fluorescence by the effectors added are interpreted as described in the text. The cyanine excitation wavelength was 470 nm, and the emission wavelength 505 nm; a.u., arbitrary units. A representative experiment is shown.

**Figure 3 jof-11-00522-f003:**
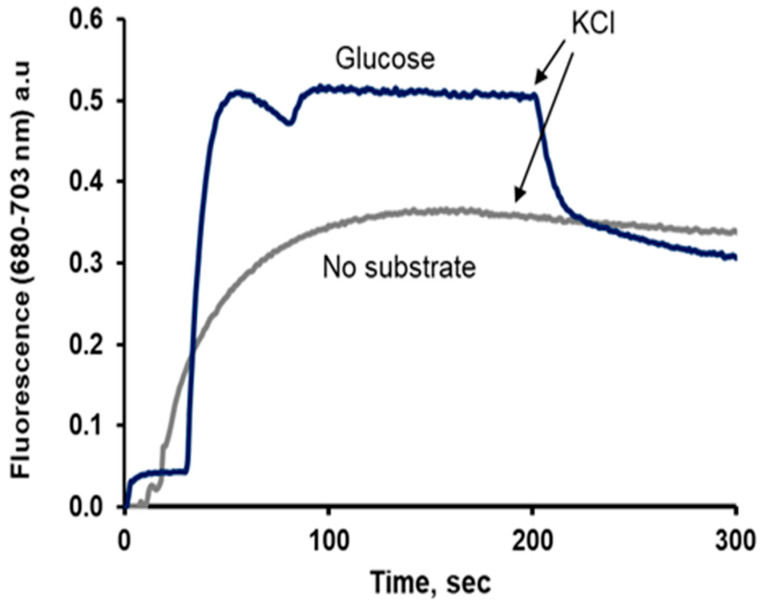
Fluorescence changes of DiSC_3_(5), added to starved *S. cerevisiae* cells at a concentration of 200 nM in the absence or presence of glucose (50 mM) and 5 µM BaCl_2_, and the effect of KCl (5 mM) added at the arrows. Excitation wavelength 680 nm, emission wavelength 703 nm; a.u., arbitrary units. A representative experiment is shown.

**Figure 4 jof-11-00522-f004:**
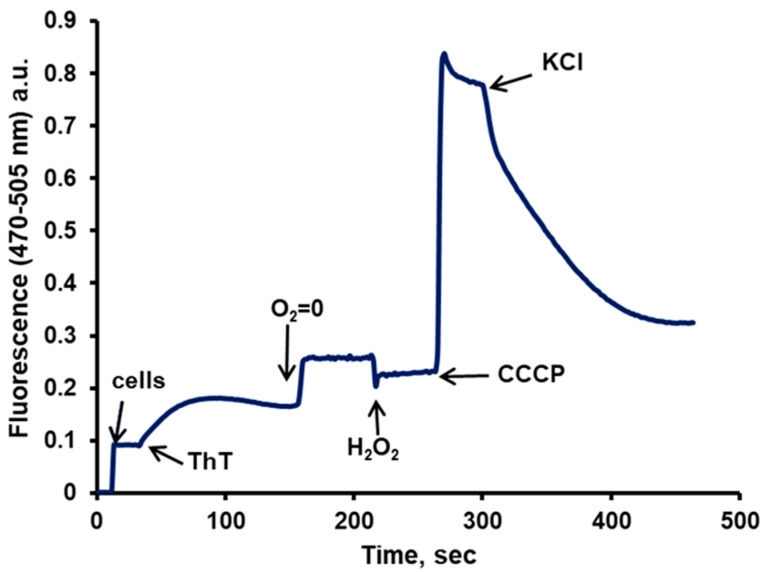
A typical fluorescence plot obtained with starved *S. cerevisiae* cells and 25 µM Thioflavin T (ThT), depicting the changes in PMP with the different effectors as explained in the text. The incubation medium contained 10 mM MES-TEA (Morpholinoethanesulfonic acid adjusted to pH 6.0 with triethanolamine) buffer, 10 µM BaCl_2_ and 20 mM glucose. Excitation wavelength 470 nm, emission wavelength 505 nm; a.u., arbitrary units.

**Figure 5 jof-11-00522-f005:**
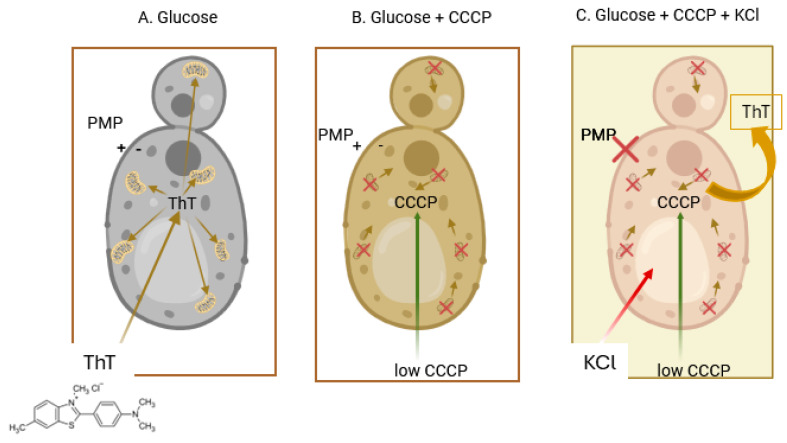
Schematic representation of the fluorescence changes of thioflavin T (ThT) by yeast cells. (**A**) In the presence of glucose, the dye enters the cytoplasm, but it is largely sequestered by the mitochondria due to their also negative potential inside; the concentration reached is so high that fluorescence is greatly quenched. (**B**) Upon the addition of a low concentration of CCCP, 10 µM, ThT leaves the mitochondria and distributes in the cytoplasm, where due to a decrease in its relative concentration, fluorescence increases. (**C**) Finally, upon the addition of KCl, ThT leaves the cell, owing to the abatement of the PMP and the total fluorescence largely decreases. Created in BioRender. SÁNCHEZ, N. S. (2025) https://BioRender.com/v28k816 (accessed on 21 January 2025).

## Data Availability

Data is available upon request. The original contributions presented in this study are included in the article and references supplied. Further inquiries can be directed to the corresponding authors.

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
