# Peer review of "From the Metabolic Effects and Mechanism of Monovalent Cation Transport to the Actual Measurement of the Plasma Membrane Potential in Yeast"

_jof, 2025, doi:10.3390/jof11070522_

Round 1

Reviewer 1 Report

1. This review provides a detailed and personal account of the development of methods to measure PMP in yeast, largely based on the authors’ long-standing work. While this historical perspective is valuable, the narrative would benefit from clearer organization. At times, the manuscript reads more like a research memoir than a concise scientific review. Consider reorganizing the content into clearer sections—such as historical background, methodological developments, comparative analysis, and current best practices—to help readers navigate the material more easily.

2. The review at times comes across as overly critical of other methods or research groups. For example, the discussion around DiSC3(3) and other dyes tends to dismiss alternative interpretations rather strongly. A more balanced tone would strengthen the review. While it’s important to point out limitations, acknowledging the contributions and context of other studies will help maintain a constructive and professional tone.

3. The manuscript reiterates many experimental details already published by the authors in past papers, especially regarding the re-identification of “acridine yellow” as Thioflavin T. While this is historically important, it occupies a disproportionate portion of the review and may distract from the broader scientific message. Condense the anecdotal content and focus more on synthesizing what current and future researchers should take away from the body of evidence surrounding PMP measurement.

4. Figures (e.g., Figures 2–5) are useful but lack sufficient standalone explanatory power. Some interpretations are only clear after extensive reading of the text. Revise figure legends for clarity and include annotations or schematics that better convey experimental designs and outcomes at a glance.

5. Some recent and relevant works on membrane potential measurements in yeast or other fungi are not cited, which could enrich the review. Include a broader range of contemporary references (2020–2024) for context, especially from groups that have applied alternative potential-sensitive probes or membrane models in yeast.

Abstract: The abstract effectively summarizes the historical context but should more clearly state the main conclusion—that ThT is currently the most reliable dye for estimating PMP when proper corrections are applied.

Line 89–93: The statement about pH effects on internal pH lacks recent references. Consider citing more modern work demonstrating yeast pH regulation under stress conditions.

Line 146–167: The interpretation of fluorescence shifts in Figure 2 is insightful but could benefit from a schematic illustration accompanying the text to visually explain dye localization.

Line 229–251: The anecdote about the misidentified dye, while interesting, is too detailed. A brief explanation with a pointer to the original publication would suffice.

Line 321–330: The conclusions are somewhat repetitive. Focus on a concise summary of recommended best practices for PMP measurements.

Line 331–339: The tribute to deceased researchers is meaningful but may be better placed in an acknowledgments or footnote section rather than the main text.

Author Response

REVIEWER 1

English language and style

The English could be improved to more clearly express the research.

Answer: The manuscript has been revised by a language expert.

Major comments

  1. This review provides a detailed and personal account of the development of methods to measure PMP in yeast, largely based on the authors’ long-standing work. While this historical perspective is valuable, the narrative would benefit from clearer organization. At times, the manuscript reads more like a research memoir than a concise scientific review. Consider reorganizing the content into clearer sections—such as historical background, methodological developments, comparative analysis, and current best practices—to help readers navigate the material more easily.

Answer: The authors thank the reviewer´s comments and suggestions. The manuscript is mainly organized based on the historical development of this theme; we change the introduction and added some new sections trying to be clearer for the readers.

  1. The review at times comes across as overly critical of other methods or research groups. For example, the discussion around DiSC3(3) and other dyes tends to dismiss alternative interpretations rather strongly. A more balanced tone would strengthen the review. While it’s important to point out limitations, acknowledging the contributions and context of other studies will help maintain a constructive and professional tone.

Answer: The reviewer is right, thank you very much for your accurate opinion. It is always best to keep things peaceful. We have included a positive comment in this respect, recognizing the value of the attempts of the German-Czech group to the advancement of the matter. However, it is clear that the conclusions of these groups were wrong, since what they observed was due to the accumulation of the dye by the mitochondria. We have, however, the text has been smoothed and included a positive note.

  1. The manuscript reiterates many experimental details already published by the authors in past papers, especially regarding the re-identification of “acridine yellow” as Thioflavin T. While this is historically important, it occupies a disproportionate portion of the review and may distract from the broader scientific message. Condense the anecdotal content and focus more on synthesizing what current and future researchers should take away from the body of evidence surrounding PMP measurement.

Answer: This part of the manuscript has been shortened.

  1. Figures (e.g., Figures 2–5) are useful but lack sufficient standalone explanatory power. Some interpretations are only clear after extensive reading of the text. Revise figure legends for clarity and include annotations or schematics that better convey experimental designs and outcomes at a glance.

Answer: Legends of the figures have been modified to specify the experimental conditions.

  1. Some recent and relevant works on membrane potential measurements in yeast or other fungi are not cited, which could enrich the review. Include a broader range of contemporary references (2020–2024) for context, especially from groups that have applied alternative potential-sensitive probes or membrane models in yeast.

Answer: We performed a new search in PubMed with the query: “Yeast membrane potential” from the last 5 years and 1460 results were obtained, many of them not precisely related to PMP (Plasma Membrane Potential) in yeast. After analyzing one by one, we opted to include 7 references that were among the more related to the scope of our review. We thank the reviewer for the suggestion, of course it enriched our work.

Detailed comments

Abstract: The abstract effectively summarizes the historical context but should more clearly state the main conclusion—that ThT is currently the most reliable dye for estimating PMP when proper corrections are applied.

Answer: We have modified the abstract in this respect

Line 89–93: The statement about pH effects on internal pH lacks recent references. Consider citing more modern work demonstrating yeast pH regulation under stress conditions.

Answer: We know that there is much modern work regarding the pH regulation in yeast as a result of stress. Yeast cells employ sophisticated mechanisms to regulate their internal pH when faced with environmental stress, particularly high or low pH conditions. These responses involve intricate signaling pathways, gene expression changes, and alterations in nutrient uptake that now have been depicted for example with RNASeq, but our purpose in the manuscript was to follow the timeline in the historical context of the discoveries that led to an understanding of the relationship between PMA1p activity and external pH and cation transport for the establishment of PMP; we apologize for leaving it that way.

Line 146–167: The interpretation of fluorescence shifts in Figure 2 is insightful but could benefit from a schematic illustration accompanying the text to visually explain dye localization.

Answer: This is explained in Figure 5 legend as it is the same with ThT.

Line 229–251: The anecdote about the misidentified dye, while interesting, is too detailed. A brief explanation with a pointer to the original publication would suffice.

Answer: We have condensed this paragraph.

Line 321–330: The conclusions are somewhat repetitive. Focus on a concise summary of recommended best practices for PMP measurements.

Answer: We have revised the conclusions

Line 331–339: The tribute to deceased researchers is meaningful but may be better placed in an acknowledgments or footnote section rather than the main text.

Answer: As reviewer 2 asked to include more on the research of our admired Mexican colleagues, we included a short paragraph at the end of the manuscript.

The authors thank the time and effort of both reviewers. Their suggestions and comments improved our work.

Reviewer 2 Report

The manuscript is suitable for publication in the Journal of Fungi, but significant revisions are required.

Point 1: It is recommended that the present mini-review  be restructured into the following sections: 

  1. Introduction
  2. Main part. In the present review, there is a section entitled 'Introduction'.
  3. Conclusions

The new 'Introduction' section should be brief. It is necessary to emphasise why this review is important and to formulate the purpose of the work, as well as its novelty and significance.

Point 2: Abbreviations should be defined the first time they appear in each of the three sections: the abstract, the main text and the first figure or table. For example, on page 85, line 85, there is an abbreviation for PMP without a transcript.

Point 3: The studies developed by José Ruiz Herrera and Aurora Brunner Liebschard are recommended to be discussed in more detail.

Point 4: The authors are encouraged to include in the review information on the works of a number of authors from Mexico working in the direction of yeast membrane potential, internal p, etc.

1) Pardo, J. P., González-Andrade, M., Allen, K., Kuroda, T., Slayman, C. L., & Rivetta, A. A structural model for facultative anion channels in an oligomeric membrane protein: the yeast TRK (K+) system // Pflügers Archiv-European Journal of Physiology 2015, 467, 2447-2460.

2) Reyes-Rosario, D.; Pardo, J.P.; Guerra-Sánchez, G.; Vázquez-Meza, H.; López-Hernández, G.; Matus-Ortega, G.; González, J.; Baeza, M.; Romero-Aguilar, L. Analysis of the Respiratory Activity in the Antarctic Yeast Rhodotorula mucilaginosa M94C9 Reveals the Presence of Respiratory Supercomplexes and Alternative Elements // Microorganisms 2024, 12, 1931. https://doi.org/10.3390/microorganisms12101931

Author Response

REVIEWER 2

Does the abstract/introduction provide a sufficiently clear description of the topic subject of this review?

No

Answer: We have included changes in the Abstract, and followed the recommendations of both reviewers, hoping the new version may be acceptable.

It is recommended that the present mini-review be restructured into the following sections: 1. Introduction 2. Main part. In the present review, there is a section entitled 'Introduction'. 3. Conclusions The new 'Introduction' section should be brief. It is necessary to emphasize why this review is important and to formulate the purpose of the work, as well as its novelty and significance.

Are the conclusions supported by results?

No

Answer: We have revised the conclusions, however, we must insist that our results are enough to support the use of thioflavin T as the best molecule, both to follow by the fluorescence changes and the corrected accumulation values, if not with all yeast species, as stated in the text, at least clearly with S. cerevisiae.

Studies developed by José Ruiz Herrera, Aurora Brunner Liebshard are recommended.

Answer: We have included what may be an adequate reference to the work of these authors.

Major comments

The manuscript is suitable for publication in the Journal of Fungi, but significant revisions are required.

Detailed comments

Point 1: It is recommended that the present mini-review be restructured into the following sections: 

  1. Introduction
  2. Main part. In the present review, there is a section entitled 'Introduction'.
  3. Conclusions

The new 'Introduction' section should be brief. It is necessary to emphasise why this review is important and to formulate the purpose of the work, as well as its novelty and significance.

Answer: We thank the reviewer the suggestions. A new introduction section emphasizing the importance of writing a review in this subject is now added and new sub-sections are added to help readers.

Point 2: Abbreviations should be defined the first time they appear in each of the three sections: the abstract, the main text and the first figure or table. For example, on page 85, line 85, there is an abbreviation for PMP without a transcript.

Answer: The PMP abbreviation is already described where the reviewer suggested.

Point 3: The studies developed by José Ruiz Herrera and Aurora Brunner Liebschard are recommended to be discussed in more detail.

Answer: We have included three references that may be examples of their work.

Point 4: The authors are encouraged to include in the review information on the works of a number of authors from Mexico working in the direction of yeast membrane potential, internal p, etc.

1) Pardo, J. P., González-Andrade, M., Allen, K., Kuroda, T., Slayman, C. L., & Rivetta, A. A structural model for facultative anion channels in an oligomeric membrane protein: the yeast TRK (K+) system // Pflügers Archiv-European Journal of Physiology 2015, 467, 2447-2460.

2) Reyes-Rosario, D.; Pardo, J.P.; Guerra-Sánchez, G.; Vázquez-Meza, H.; López-Hernández, G.; Matus-Ortega, G.; González, J.; Baeza, M.; Romero-Aguilar, L. Analysis of the Respiratory Activity in the Antarctic Yeast Rhodotorula mucilaginosa M94C9 Reveals the Presence of Respiratory Supercomplexes and Alternative Elements // Microorganisms 2024, 12, 1931. https://doi.org/10.3390/microorganisms12101931

Answer: These references, and the pertinent comments have been included.

The authors thank the time and effort of both reviewers. Their suggestions and comments improved our work.

Round 2

Reviewer 1 Report

This revised mini-review offers a valuable and thorough overview of the historical development and modern methodologies for estimating plasma membrane potential in yeast, with a particular emphasis on fluorescent dyes and genetically encoded indicators. The narrative effectively integrates the authors’ long-standing contributions to the field, contextualized within broader scientific progress.

However, while the scientific content is strong, the clarity of expression could be improved in many areas. Several paragraphs are overly long, and certain sections could benefit from clearer organization, simplified phrasing, and more polished grammar. Improving the English language quality would significantly enhance readability and impact.

  1. Introduction: It would be clearer if the last sentence of the paragraph were streamlined to avoid repetition (“helping the scientific community make new discoveries...” could be shortened or rephrased). Suggest clarifying the importance of accurate PMP measurement by briefly stating why prior results were inconsistent.
  2. line 56: "monovalent action" should be corrected to "monovalent cation"
  3. 3.1 – Dye-based methods: Consider briefly summarizing the limitations of each dye after its discussion to enhance clarity for readers unfamiliar with their mechanisms.
  4. Conclusions: You may consider adding 1–2 lines suggesting potential future directions (e.g., broader application of GEVI proteins, relevance in industrial or stress conditions).

Author Response

Response to Reviewer 1 Comments

1. Summary

Thank you very much for taking the time to review this manuscript. Please find the detailed responses below and the corresponding revisions/corrections highlighted/in track changes in the re-submitted file.

2. Questions for General Evaluation

Reviewer’s Evaluation

Response and Revisions

Does the introduction provide sufficient background and include all relevant references?

Yes/

We thank the reviewer

Are all the cited references relevant to the research?

Yes/

Is the research design appropriate?

Yes/

Are the methods adequately described?

Yes/

Are the results clearly presented?

Yes/

Are the conclusions supported by the results?

Yes/

3. Point-by-point response to Comments and Suggestions for Authors

Major comments: This revised mini-review offers a valuable and thorough overview of the historical development and modern methodologies for estimating plasma membrane potential in yeast, with a particular emphasis on fluorescent dyes and genetically encoded indicators. The narrative effectively integrates the authors’ long-standing contributions to the field, contextualized within broader scientific progress.

However, while the scientific content is strong, the clarity of expression could be improved in many areas. Several paragraphs are overly long, and certain sections could benefit from clearer organization, simplified phrasing, and more polished grammar. Improving the English language quality would significantly enhance readability and impact.

Response to Major comments: Thank you for your nice comments, and thank you also for the suggestions. We have accordingly made certain changes in the grammar and the length of the paragraphs, also modifying the organization of the sections to make an easier readability of the review. We hope that the new version will be satisfactory.

Detailed comments:

  1. Introduction: It would be clearer if the last sentence of the paragraph were streamlined to avoid repetition (“helping the scientific community make new discoveries...” could be shortened or rephrased). Suggest clarifying the importance of accurate PMP measurement by briefly stating why prior results were inconsistent.

Response to detailed comments 1: We have revised the introduction to avoid repetition and follow your suggestions.

  1. line 56: "monovalent action" should be corrected to "monovalent cation"

Response to detailed comments 2: We are sorry for the mistake. It has been changed.

  1. 3.1 – Dye-based methods: Consider briefly summarizing the limitations of each dye after its discussion to enhance clarity for readers unfamiliar with their mechanisms.

Response to detailed comments 3: We tried to include this suggestion all along section 3.1 since the previous version. We hope that with the new organization, it is clearer for readers.

  1. Conclusions: You may consider adding 1–2 lines suggesting potential future directions (e.g., broader application of GEVI proteins, relevance in industrial or stress conditions).

Response to detailed comments 4: Your suggestion is now included, thank you.

5. Additional clarifications

Finally, we want to thank Reviewer1 for his/her conscientious review of our manuscript and the time invested on it. This is truly valuable.

Reviewer 2 Report

The corrected version of the article can be published in JoF.

The corrected version of the article can be published in JoF.

Author Response

The authors wish to thank Reviewer 2 for the time invested in the review of our manuscript.